# Text-guided RGB-P grasp generation

Van Duc Vu[1], Van Thiep Nguyen[1], Nam Hai Pham[1],
Dinh-Cuong Hoang[1] and Phan Xuan Tan[2]

[1] IT Department, FPT University, Ha Noi, Vietnam
[2] College of Engineering, Shibaura Institute of Technology, Tokyo, Japan

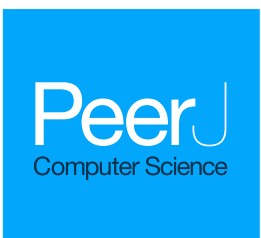

## ABSTRACT

In the field of robotics, object grasping is a complex and challenging task. Although state-of-the-art computer vision-based models have made significant progress in predicting grasps, the lack of semantic information from textual data makes them susceptible to ambiguities in object recognition. For example, when asked to grasp a specific object on a table with many objects, robots relying only on visual data can easily get confused and grasp the wrong object. To address this limitation, we propose a multimodal approach that seamlessly integrates 3D data (shape) and red-green-blue (RGB) images (color, texture) into a unified representation called red-green-blue and point cloud (RGB-P), while also incorporating semantic information from textual descriptions processed by a large language model (LLM) to enhance object disambiguation. This combination of data allows our model to accurately infer and capture target objects based on natural language descriptions, overcoming the limitations of vision-only approaches. Our approach achieves superior performance, with an average precision (AP) of 53.2% on the GraspNet-1Billion dataset, significantly outperforming state-of-the-art methods. Additionally, we introduce an automated dataset creation pipeline that addresses the challenges of data collection and annotation. This pipeline leverages cutting-edge models: LLMs for text generation, Stable Diffusion for image synthesis, Depth Anything for depth estimation, using standard intrinsic parameters from the Kinect depth sensor to ensure geometric consistency, and GraspNet for grasp estimation. This automated process generates high-quality datasets with paired RGB-P, images, textual descriptions and potential grasp poses, significantly reducing the manual effort and enabling large-scale data collection.

## INTRODUCTION

Human-robot communication in grasp detection acts as a supplementary source of information, helping robots better understand task requirements and perform precise grasping actions (*Mees et al., 2022*). This is particularly useful in cluttered environments, where the ability to communicate and learn from humans aids in creating focused grasps on target objects. For example, consider a scenario with two apples and various other objects on a table; determining how to grasp the two apples involves addressing two main challenges: (1) identifying suitable grasping points and (2) recognizing the object to be grasped. Grasp detection research can be categorized into two main approaches, including planar grasping and six-degrees-of-freedom (6-DoF) grasping. Planar grasping

Corresponding author
Phan Xuan Tan,
tanpx@shibaura-it.ac.jp

(*Jiang, Moseson & Saxena, 2011*; *Lenz, Lee & Saxena, 2015*; *Chu, Xu & Vela, 2018*; *Guo et al., 2017*) based on simpler grasp representations. In contrast, 6-DoF grasping is more adept and suitable for handling complex situations. Notable studies (*Chen et al., 2023*; *Mousavian, Eppner & Fox, 2019*; *Sundermeyer et al., 2021*; *Hoang, Stork & Stoyanov, 2022*), although they achieved good grasping performance, heavily depend on dense 3D information, making them vulnerable when input sensors are unstable. To address this issue, *Zhai et al. (2023)* and their previous studies (*Xuan Tan et al., 2024*) reduced the reliance on dense 3D input by using the red-green-blue (RGB) input. These studies, which only used visual data, achieved good results but only addressed the first challenge due to the ambiguity of grasping without object recognition. However, to deal with the second challenge, understanding objects is necessary, potentially through learning from humans *via* text data.

In this study, our method seamlessly integrates 3D point cloud data (shape), RGB images (color and texture), and textual descriptions of objects for better grasp detection performance. The novelty of our approach lies in the synergistic integration of these modalities into a unified "RGB-P" representation, with textual descriptions processed by a large language model (LLM) to provide semantic context, enabling precise object identification and grasp generation in cluttered environments. The proposed method processes text input through a text encoder to exploit language interpretability for a deeper understanding of the object, while visual data, which is a concatenation of two types of input: RGB image and Pointcloud are processed by a backbone (*Qi et al., 2017*) network to extract multi level features and maintain invariance to spatial transformations. These information streams are then combined using cosine similarity and passed through the model to generate the grasp.

Beyond the core grasping module, we introduce an automated dataset creation pipeline to address data collection and annotation challenges. In fact, creating datasets for robotic grasping is notoriously difficult, requiring the capture of various objects in complex real-world environments and accurate annotation with detailed grasp information. Additionally, these tasks are also time consuming, requiring specialized expertise, and can be expensive to scale. Our pipeline tackles these challenges by leveraging advanced technologies like LLMs and image generation models to create realistic and diverse data with reduced human effort. Specifically, LLMs (*Brown et al., 2020*) are used for text generation, Stable Diffusion (*Hu et al., 2021*) for image synthesis, Depth Anything (*Yang et al., 2024a*) for depth estimation and GraspNet (*Fang et al., 2020*) for grasp estimation, providing labeled data for potential grasp poses. This integrated approach allows us to efficiently create a large dataset tailored for training robust grasping systems. By leveraging standard intrinsic parameters from the Microsoft Kinect depth sensor, we ensure geometric consistency in the generated 3D point clouds. This automated process efficiently generates high-quality datasets with paired RGB-P, images, textual descriptions, and potential grasp poses. This significantly reduces manual effort and enables large-scale data collection, paving the way for more robust and versatile grasping systems.

In summary, our main contributions are as follows:

1. A novel text-guided grasp detection approach that integrates RGB images, 3D point clouds, and textual descriptions processed by an LLM, achieving state-of-the-art performance with an AP of 53.2% on the GraspNet-1Billion dataset.
2. An automated dataset creation pipeline that leverages advanced technologies (LLMs, Stable Diffusion, Depth Anything, GraspNet) to generate diverse, geometrically consistent grasping data with minimal human intervention.
3. We contribute a text prompt dataset describing 52 object categories from the GraspNet 1-Billion dataset.

The remainder of this article is organized as follows: Related Work and Background, provides an overview of existing research in grasp detection, including planar and 6-DoF grasping, as well as relevant studies on language models, image generation, and depth estimation. In Methodology, we introduce our proposed grasp generation pipeline ("Grasp generation pipeline"), detailing the steps involved in automated dataset creation and language-driven RGB-P grasp detection ("Language-driven RGB-P grasp detection"). This includes a description of the attention mechanism used to fuse text and visual features. Evaluation, presents the evaluation metrics ("Evaluation metrics"), comparing our approach to the state-of-the-art methods in the GraspNet-1 Billion dataset ("Evaluation on GraspNet-1Billion dataset") and the experimental results ("Grasping robot simulation experiment"). Implementation Details provides information on the practical aspects of the system, including hardware and software requirements, training parameters, and optimization techniques. Limitations discusses the constraints and challenges of our approach. Finally, Conclusions summarizes the key contributions of our work, discusses its limitations and outlines potential avenues for future research. The detailed abbreviations and definitions used in the article are listed in Table 1.

## RELATED WORK AND BACKGROUND

### Grasp detection

In this section, we will focus on the analysis of grasping methods, including 2D planar grasping and 6-DoF grasping, along with recent prominent research in this area. 2D planar grasping simplifies the task by restricting the gripper to move perpendicular to the camera plane. The oriented rectangle becomes a powerful tool to present the grasp in this plane, conveying essential information about the grasp position, orientation, and gripper opening. *Jiang, Moseson & Saxena (2011)*, combining the "grasping rectangle" representation with a two-step learning algorithm, offer a significant advancement in robot grasping capabilities for novel objects. *Lenz, Lee & Saxena (2015)* demonstrate that deep learning, coupled with a novel multimodal regularization technique, provides a powerful and effective approach for robotic grasp detection in real-world scenarios. *Chu, Xu & Vela (2018)* proposed combining grasp region proposals with classification-based orientation estimation, leading to robust and efficient grasp detection. *Guo et al. (2017)* propose a

**Table 1** List of abbreviation and acronyms.

| Abbreviation | Definition |
| --- | --- |
| RGB | Red-Green-Blue |
| RGB-D | Red-Green-Blue and depth |
| RGB-P | Red-Green-Blue and point cloud |
| 2D | Two-dimensional |
| 3D | Three-dimensional |
| DoF | Degrees of freedom |
| VAE | Variational autoencode |
| LLMs | Large language models |
| NLP | Natural language processing |
| ReLU | Rectified linear unit |
| Concat | Concatnate |
| CoS | Cosine similarity |
| Mul | Multiply matrix |
| AEM | Attention elementwise mask |
| AM | Attention mask |

novel hybrid deep learning architecture for robotic grasp detection that integrates visual and tactile sensing data. Some other notable studies include (*Watson, Hughes & Iida, 2017*; *Wang et al., 2016*; *Morrison, Corke & Leitner, 2018*). However, grasp generation is limited to the 2D plane due to the limited degrees of freedom in grasping poses, causing important grasping poses to be overlooked.

On the other hand, Six-Degrees of Freedom (6-DoF) grasping capability demonstrates the robot's finesse and flexibility in manipulating and grasping objects, similar to the human hand. It allows grasping objects from a variety of angles and directions, providing maximum flexibility in complex situations (*Chen et al., 2023*; *Zhai et al., 2023*; *Mousavian, Eppner & Fox, 2019*; *Sundermeyer et al., 2021*; *Hoang, Stork & Stoyanov, 2022*). *Chen et al. (2023)* present a highly efficient and accurate solution for 6-DoF grasp pose detection. By leveraging heatmaps for guidance and incorporating semantic information, it effectively addresses the challenges of grasping in cluttered scenes. *Mousavian, Eppner & Fox (2019)* proposed a new method called 6-DoF GraspNet to generate diverse and successful grasps for unknown objects. This method consists of two approaches including using a variational autoencoder (VAE) to generate diverse grasp poses from the object's point cloud, and using a neural network to evaluate the grasp ability, assigning a probability of success to each pose. Contact-GraspNet (*Sundermeyer et al., 2021*) is an end-to-end neural network to generate 6-DoF grasp poses for unknown objects in cluttered scenes, directly from a depth image. *Hoang, Stork & Stoyanov (2022)* took advantage of a voting mechanism and contextual information to directly generate grasp configurations from 3D point clouds, solving common occlusion challenges in manipulation.

Although the above studies have achieved good results, there are still some problems. The accurate generation of 6-DoF grasps often requires geometric information, making many existing methods dependent on 3D point cloud data. In addition, RGB information

can provide additional contextual information about the texture, color, and shape of the object, which can improve the robustness and reliability of grasp detection algorithms. Contributing to the solution of the 6-DoF problem, our previous work (*Xuan Tan et al., 2024*) proposes a novel deep learning framework named Attention-based Grasp Detection with Monocular Depth Estimation for 6-DoF grasp generation using only an RGB image as input. *Zhai et al. (2023)* propose MonoGraspNet, the first deep learning pipeline for 6-DoF robotic grasping using only a single RGB image. This is in contrast to previous methods that heavily rely on depth sensors, which perform poorly on objects with challenging photometric properties such as transparency or reflectivity. However, some limitations, such as relying solely on visual data, lead to overlooking an important aspect of human-robot interaction. New dataset such as *Vuong et al. (2024)*, which combine visual and textual data, show that combining text can help to create better grasps. Our method seamlessly integrates 3D point cloud data (shape), RGB images (color and texture), and text descriptions of objects into a unified "RGB-P". Specifically, we use a state-of-the-art LLM to capture and model the grasp generation process, enhancing human-robot interactions. This method enables the generation of object-specific, object-centered grasps based on text descriptions, providing greater accuracy and flexibility in real-world scenarios.

## LLMs-large language models

Natural language processing (NLP) was born as an ambitious human effort to bridge the gap between two seemingly separate worlds, the world of natural language and the digital world. Currently, its application in everyday life is increasing due to its practical benefits in life thanks to its large language model. Studies that have had a major impact on development, such as *Vaswani et al. (2017)* introduced the Transformer architecture, which has become the backbone of numerous state-of-the-art models in NLP and beyond, including BERT and GPT-3. *Peters et al. (2018)* proposed ELMo (Embeddings from Language Models), an innovative method to generate deep, contextualized word representations. ELMo employs a bidirectional long short-term memory language model (biLSTM), pre-trained on an extensive text *corpus*, to capture intricate word characteristics (syntax and semantics) and their variations across different contexts (polysemy). *Radford & Narasimhan (2018)* developed a semi-supervised approach to language understanding tasks, focus on the use of a generative pre-trained language model for various downstream applications. *Devlin et al. (2019b)* introduced Bidirectional Encoder Representations from Transformers (BERT), a groundbreaking technique for pre-training language representations that addresses the limitations of earlier models like ELMo and OpenAI GPT by enabling deep bidirectional training. The emergence of LLMs such as Gemini (*Team et al., 2024*) developed by Google DeepMind represents a new era in natural language processing. As a next-generation multimodal LLM, Gemini stands out with its capability to process ultra-long contexts of up to 10 million tokens and retrieve information with exceptional accuracy (>99%) across diverse data modalities, including text, video, and audio. Alongside other cutting-edge LLMs (*Brown et al., 2020*; *Touvron et al., 2023*; *Chowdhery et al., 2022*; *Achiam et al., 2023*), these advances have marked a

major leap forward in the field, allowing the generation of creative and high-quality content across a wide range of genres, from poetry, short stories, and screenplays to articles, emails, and social media post, with a level of fluency and coherence that closely mirrors human writing.

## LLM-based robotic grasping

Recent advancements in LLMs have significantly influenced robotic grasping by enabling semantic understanding and task-oriented manipulation. In which *Jin et al. (2024)* proposed a new task called reasoning grasping, where a robot performs reasoning to determine grasping poses based on implicit instructions. To accomplish this task, they developed a system that integrates a multimodal large language model—specifically LLaVA—that enables simultaneous processing of text and RGB image data to extract information for object recognition and manipulation. However, using only RGB and text data as model inputs have certain limitations. Specifically, in scenarios with multiple overlapping objects, RGB images may not provide sufficient information because the target object is occluded. This leads to conflicts between the image and text description, causing the linguistic information to no longer accurately reflect the characteristics of the object to be manipulated. *Wang et al. (2024)* proposed the Polaris framework, using the power of GPT-4 to analyze natural language instructions and extract target queries. These queries are then integrated with the RGB-D image data through a vision module, enabling the system to perform robot control tasks, such as object grasping, effectively. Meanwhile, *Tang et al. (2023)* introduced an innovative modular architecture for task-oriented grasping (TOG). This framework uses a LLM to generate open-ended semantic descriptions, allowing the model to generalize in a zero-shot manner to objects and tasks not present in the training set. GraspGPT serves as a grasp pose evaluator, receiving grasp candidates from Contact-GraspNet, a pre-trained sampling module. However, a notable limitation of both approaches is their complete reliance on linguistic input without using visual information during the query semantics determination phase. This leads to difficulties in handling referentially ambiguous instructions. For example, in a scenario with two cups of different colors (one blue and one white) on a table, a simple instruction like "pick up the cup" may prevent the model from identifying the target object. This ambiguity reduces the accuracy and automation level of the system, particularly in real-world environments with multiple similar objects. *Xu et al. (2023)* presented a method integrating vision, language, and action to perform target object grasping in cluttered environments, using deep reinforcement learning with the Soft Actor-Critic algorithm and pre-trained models like CLIP and GraspNet. However, this method has limitations in handling complex linguistic instructions. The system relies on fixed language templates such as "Give me the {keyword}" or "Grasp a {keyword} object," reducing flexibility when processing diverse sentence structures or free-form language in real-world settings. Our work addresses these limitations by using Gemini 2.0 to generate detailed 3D object descriptions and the BERT encoder to extract robust features from the object descriptions.

## Stable diffusion

Stable Diffusion is a powerful text-to-image generation model (*Rombach et al., 2022*) that forms the core component of our automated dataset generation process. We leverage the SDXL model (*Rombach et al., 2022*) as the foundation, enhancing it with low-rank adaptation (LoRA) (*Hu et al., 2021*) to fine-tune its performance for our specific application. This adaptation process allows us to tailor the model's capabilities to generate images that are particularly relevant and useful for training our grasp generation model. The text prompts generated by Gemini (Gemini 2.0) (*Team et al., 2024*) serve as the input to Stable Diffusion, guiding the model in creating high-quality, realistic images that depict the specified scene. These images capture a wide range of variations in object arrangements, lighting conditions, and perspectives, ensuring that our training dataset is rich and diverse. This diversity is essential for our grasp generation model to learn to handle real-world scenarios where objects may be positioned in various ways, under different lighting conditions, and from different viewpoints. The emergence of text-to-image generation models has revolutionized computer vision and artificial intelligence. Models like DALL-E2 (*Ramesh et al., 2022*) and Imagen (*Saharia et al., 2022*) have also demonstrated remarkable abilities in creating photorealistic images from textual descriptions. However, Stable Diffusion stands out for its open-source nature and flexibility, making it particularly suitable for our application. The open-source nature of Stable Diffusion allows researchers and developers to freely experiment with the model and adapt it to their specific needs. Its flexibility enables us to customize the generation process by controlling various parameters, including the image resolution, the number of steps in the diffusion process, and the specific prompt engineering techniques employed. In contrast to other text-to-image models that rely on proprietary systems, Stable Diffusion allows us to fine-tune and control the dataset generation process to ensure that it aligns perfectly with our requirements for training a robust grasp generation model. This control over the dataset is crucial for enabling our robot to learn to perform grasping tasks with high accuracy and reliability in diverse real-world scenarios.

## Fusion mechanism

The fusion of textual and visual features poses a unique challenge in multimodal learning for different tasks. *Lu et al. (2022)* used different late fusion strategies for image-text multimodal classification. However, the drawback of late fusion is that it may miss capturing the complex relationships and dependencies between textual and visual features. *Yu et al. (2023)* proposed DFM, which aims to facilitate the information exchange between linguistic and visual features, helping the model learn richer and more discrete representations, ultimately leading to better component-free learning. Although DFM aims to enhance visual feature learning through guidance from language, it still relies heavily on pre-trained linguistic features. The ability of the model to infer unseen components may be limited by the expressiveness and scope of the language embedding space. *Liang et al. (2022)* introduced cosine similarity to measure how well the generated

text matches the input image. CLIP encodes the image and text into separate vectors, and the cosine similarity is calculated between these vectors. A higher cosine similarity indicates that the text is more relevant to the image, which serves as a reward signal for the reinforcement learning algorithm to learn better alignments. In this study, we merge text and image features using cosine similarity to focus on each relationship between text and image.

### Depth estimation

Accurate depth information is crucial for our language-driven RGB-D grasp generation system, as it provides the necessary three-dimensional context for effective grasp planning. Depth estimation aims to infer the distance of objects in a scene from a single or multiple images (*Yang et al., 2024a*). This information is essential for robots to accurately perceive the environment and plan their actions, especially when grasping objects. To generate depth maps for our dataset, we leverage the Depth Anything V2 algorithm (*Yang et al., 2024b*), an advanced monocular depth estimation model known for its superior accuracy and robustness.

Depth estimation is a rapidly evolving field, with numerous methods and models being developed. Other notable approaches include stereo vision (*Scharstein & Szeliski, 2003*), structured light (*Zhang, 2000*), and time-of-flight sensors (*Jähne, 2002*). However, monocular depth estimation methods like Depth Anything V2 are particularly valuable for robotics applications, as they offer a cost-effective and flexible solution for depth perception, with improved generalization to diverse real-world scenarios.

## MATERIALS AND METHODS

### Materials

This section details the dataset and codes employed in this study.

1. Code repository: The code utilized for data processing, model training, and evaluation is publicly available on Zenodo (DOI: 10.5281/zenodo.14038077). This repository includes the code itself, comprehensive installation instructions, and detailed documentation, ensuring the reproducibility of our experiments. Access the code here: https://zenodo.org/records/14038077.

2. 3rd-party dataset DOI/URL: The publicly available GraspNet dataset (https://github.com/graspnet) provided a comprehensive collection of data for object grasping tasks.

Training data: The GraspNet training data consist of four compressed files (train_1.zip to train_4.zip) downloadable directly from the following Zenodo link:

- https://zenodo.org/records/16006850

Testing data: The testing data encompasses three separate zipped archives: test_seen.zip, test_similar.zip, and test_novel.zip downloadable directly from the following Zenodo link:

- https://zenodo.org/records/16006850

Additionally, separate label files (grasp_label.zip and collision_label.zip) provide the corresponding ground-truth labels. These files are available at the following Zenodo link:

- https://zenodo.org/records/16006850

3. Data extraction steps

To prepare the data for model training and testing, you need to carefully follow the steps below to ensure that the data are properly organized and ready for use. First, download the GraspNet dataset directly from the following Google Drive links above to download the following files: `train_1.zip`, `train_2.zip`, `train_3.zip`, `train_4.zip`, `grasp_label.zip`, `collision_grasp.zip`, `test_seen.zip`, `test_similar.zip`, and `test_novel.zip`. After downloading, create a root directory named `data` to store all the data. Within the `data` directory, create four subdirectories with the following organization and processing:

- Directories `grasp_label` and `collision_grasp`:

Extract the files `grasp_label.zip` and `collision_grasp.zip`. After extraction, you will have two folders named `grasp_label` and `collision_grasp`.

- Directory `scenes`:

Extract all the files `train_1.zip`, `train_2.zip`, `train_3.zip`, `train_4.zip`, `test_seen.zip`, `test_similar.zip`, and `test_novel.zip`. From the extracted files, take all the subdirectories and consolidate them into a single folder named `scenes`.

- Directory `text data`:

This research utilizes text data provided in Supplemental File 1. After downloading and extracting, you will have a folder named `text_data` inside the `data` directory.

Next, you need to generate tolerance labels by using the script `generate_tolerance_label.py`. This script requires the path to the `data` folder as input. Make sure to provide the correct path and execute the script. Finally, you can begin the training or testing process by running the files `train.py` or `test.py`. Before running these scripts, ensure that all the paths within the scripts are configured correctly to point to the `data` directory with the structure outlined above.

To streamline data handling and ensure efficient model training, the downloaded data have been meticulously extracted and organized into a structured directory as follows:

```
root:
  data
    grasp_label
      folder containing label data from grasp_label.zip
    collision_label
      folder containing label data from collision_label.zip
    scenes
      folder containing extracted scene data (train_1.zip to train_4.zip,
      test_seen.zip, test_similar.zip, and test_novel.zip) from GraspNet
    text_data
      folder containing the text dataset, including data.json and vocab.
      json.
```

## Methods

The raw data was considered sufficiently clean and suitable for the intended analysis. Therefore, no data pre-processing steps such as cleaning, transformation, or feature engineering were applied.

## Synthetic data generation details

The generated image is processed using a depth estimate model, such as Depth Anything V2 (*Yang et al., 2024b*), to extract depth information. For converting these relative depth maps into 3D point clouds suitable for GraspNet (*Fang et al., 2020*), we utilize standard intrinsic parameters from the Kinect depth sensor. Our automated pipeline generates a large volume of synthetic RGB images using Stable Diffusion XL. From this pool, we select images where the projection of the Depth Anything V2 output, enhanced by its teacher-student framework for improved metric accuracy, results in coherent 3D geometry and yields high-confidence grasp proposals from the subsequent GraspNet annotation stage. This selection ensures that the generated 3D data are well-formed and aligns closely with real-world sensor characteristics.

## Grasp generation pipeline

Our grasp generation pipeline is a multistage process that leverages a combination of language processing, image generation, and depth estimation to create a robust dataset to train a grasp generation model. This pipeline efficiently automates the process of creating a diverse and realistic dataset for robust grasp generation. See Fig. 1 for a visual representation of the pipeline.

### *Language-based scene description*

The pipeline begins with a textual description provided by the user of a scene, acting as a natural language instruction. For example, a user might input "Describe a bottle, an orange, and an apple on a table to put into Stable Diffusion." This text provides a high-level representation of the desired scene for dataset generation (*Devlin et al., 2019a*).

### *Prompt generation with LLMs*

The textual description is processed by a LLM (*Radford et al., 2019*; *Brown et al., 2020*), such as Gemini (*Team et al., 2024*), to generate a detailed text prompt suitable for image generation. The LLM analyzes the input text, comprehending the object, their attributes, and their relationship with the scene. Based on this understanding, the LLM constructs a comprehensive text prompt incorporating descriptions of objects, their relative positions, potential lighting conditions, and other relevant attributes relevant to grasp generation.

### *Text generation on available GraspNet data*

The availability of comprehensive textual descriptions is crucial for effectively utilizing GraspNet data in language-driven grasp generation. However, for existing GraspNet datasets lacking such descriptions, the manual process of creating them can be incredibly time consuming and laborious. To address this challenge, we propose an efficient and scalable solution: taking advantage of LLMs (*Radford et al., 2019*; *Brown et al., 2020*) to

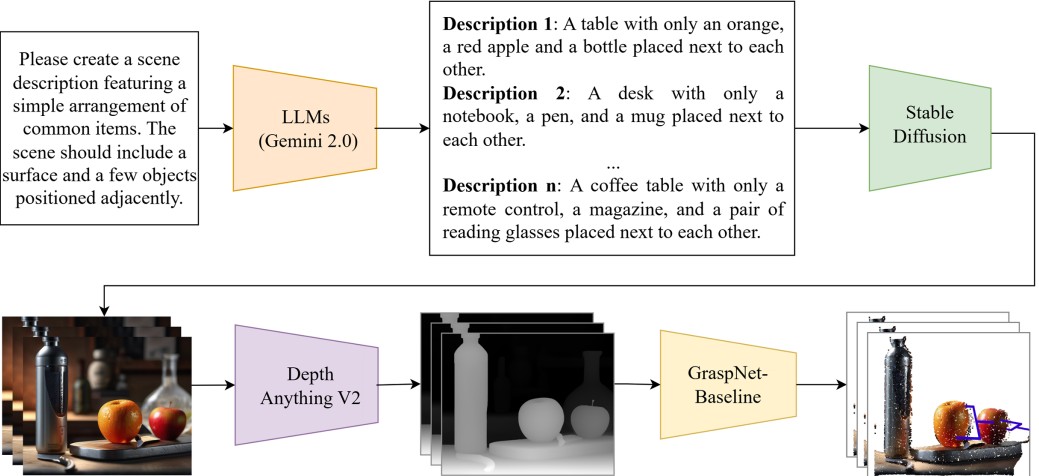

**Figure 1 Automated dataset generation pipeline.** This pipeline creates dataset using language-based scene descriptions, LLMs for prompt generation, stable diffusion for image synthesis, depth anything for depth estimation, and GraspNet for grasp pose estimation.

automatically generate text descriptions for the scene and objects depicted in the data. LLMs are capable of analyzing the available GraspNet data, understanding the objects present, their attributes, and their spatial relationships within the scene. This understanding enables them to construct comprehensive text prompts that not only describe the objects themselves but also incorporate details such as their relative positions, potential lighting conditions, and other relevant attributes. These attributes are critical for generating accurate and contextually relevant grasps. By automating the process of text description generation, we significantly reduce the manual effort required to enrich existing GraspNet datasets. This enables us to expand the scope of language-driven grasp generation, allowing us to train more robust and versatile grasping systems using a wider range of data (See Fig. 2 for examples of text descriptions.).

### Image generation with stable diffusion

The generated text prompt is fed into Stable Diffusion (*Rombach et al., 2022*), a powerful text-to-image generation model, which interprets the text prompt and produces a realistic image depicting the scene described by the user. We utilize SDXL (Stable Diffusion XL) as the base model and enhance it with LoRA (*Hu et al., 2021*) (Low-Rank Adaptation) for improved performance in our specific application. This image serves as a visually accurate representation of the scene, capturing the specified objects, their arrangements, and relevant environmental details.

### Depth estimation for grasp planning

The generated image is then processed by the Depth Anything V2 model (*Yang et al., 2024b*), an advanced depth estimation model, to extract depth information. This process analyzes the image and estimates the depth value for each pixel, resulting in a depth map that represents the scene. The depth map provides critical spatial information for the grasp generation model, enabling it to accurately understand the

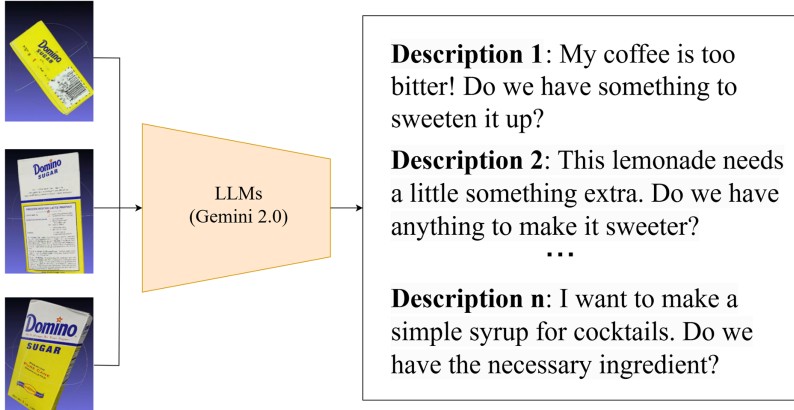

**Figure 2  Example text descriptions for ingredient identification.**

relative positions and orientations of objects in the scene. This step is crucial for the model to effectively understand the scene and plan grasps. Depth Anything V2 leverages a teacher-student framework, where a teacher model trained on synthetic data generates pseudo-labels for real-world images, enhancing metric accuracy and reducing the domain gap, thus improving the quality of the generated depth maps for robotic applications.

## Grasp generation and training

The generated image and its corresponding depth map are transformed into a 3D point cloud and then fed into GraspNet (*Fang et al., 2020*). This model analyzes the scene, using both visual and depth information, to identify potential grasp points for a robotic manipulator. The model outputs a set of candidate grasp poses, providing the robot with the information needed to safely and effectively grasp the desired objects.

### Language-driven RGB-P grasp detection

The aim of our proposal is to estimate grasps for objects described in text combined with RGB images and point clouds. The proposed method is illustrated in Fig. 3. The features are first extracted from text, RGB image, and point cloud using the corresponding backbone. Attention weights are then calculated between the features from different sources using cosine similarity. Finally, the attended features are passed through a text-guided grasp generation module, refined based on GraspNet (*Fang et al., 2020*), a baseline model, to estimate grasps for the object.

### RGB-P

Based on the Pointnet++ mechanism, we concatenated the point cloud with the RGB image to form an RGB-P map. This combined map provides the model with a richer understanding of the scene, which includes both geometric and visual information. Using the RGB-P map, PointNet++ learns more comprehensive features for keypoint selection,

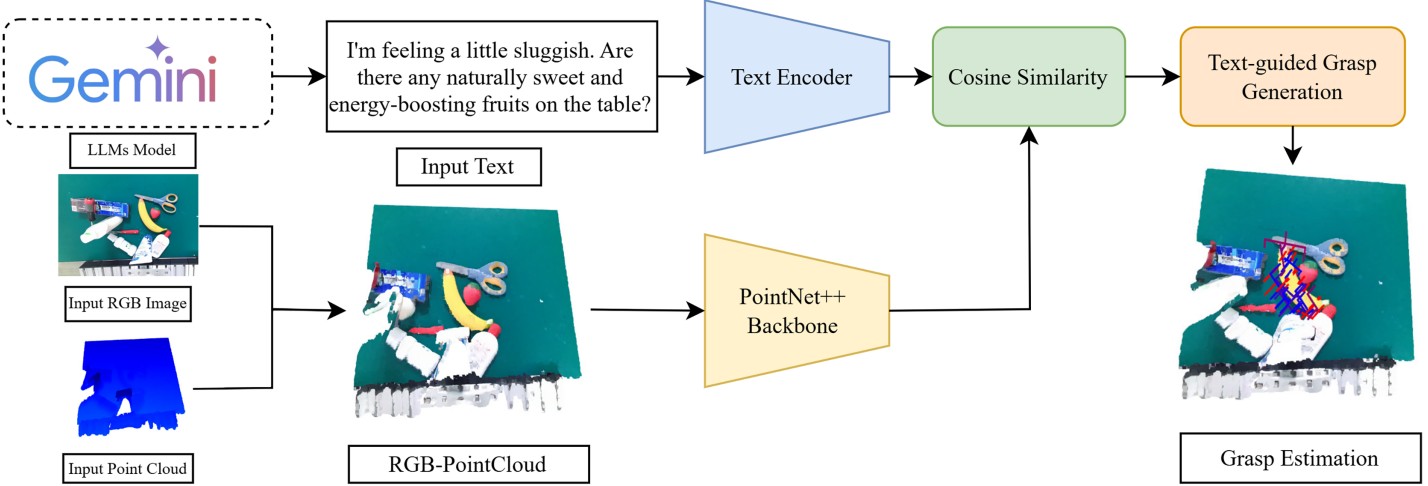

**Figure 3 Model architecture: this diagram includes the modules of our network, a text encoder (BERT (*Devlin et al., 2019a*) model), 3D backbone (PointNet++ (*Qi et al., 2017*)), attention mechanism based on cosine similarity matrix (*Lee et al., 2018*), text-guided grasp generation (*Fang et al., 2020*).**

leading to improved performance in tasks such as object classification, segmentation, and registration.

### Backbone

Given an input text description, an RGB image, and a point cloud. After combining the RGB image and the point cloud into an RGB-P map, we use the corresponding models to extract features, denoted $F_{text}$ from a text and $F_{rgbp}$ from an RGB-P map. The textual features $F_{text} \in \mathbb{R}^{L \times 256}$ are extracted from the text using the backbone module of the BERT model (*Devlin et al., 2019a*), where $L$ denotes the length of the text. In addition, spatial and appearance features extracted from RGB-P map representation $P = \{p_1, \ldots, p_n\} \in \mathbb{R}^6$ combine the RGB image and the depth image transformed to the 3D pointcloud, using Pointnet++ backbone, which denotes $F_{rgbp} \in \mathbb{R}^{N \times 256}$.

### Attention mechanism with cosine similarity

Taking as input textual features $F_{text} \in \mathbb{R}^{L \times 256}$ and spatial, appearance features $F_{rgbp} \in \mathbb{R}^{N \times 256}$, this module implements an attention mechanism based on cosine similarity (*Lee et al., 2018*), to measure the similarity between textual features and each point in the point cloud features, is illustrated in Fig. 4.

$F_{rgbp}$ and $F_{text}$ will calculate the cosine similarity matrix $\in \mathbb{R}^{N \times L}$. This matrix measures the similarity between each pair of features. The *AEM* (*AttentionElement-wiseMask*) is generated from the cosine similarity matrix, high similarity represents high attention and vice versa. This mask indicates which features in $F_{text}$ should be attended to for each feature in $F_{rgbp}$, helps to focus on important features in $F_{text}$.

$F_{text}$ is multiplied by *AEM*. This produces the *AM* (*AttentionMask*) representation $AM = \{am_1, \ldots, am_n\} \in \mathbb{R}^{256}$, shows the correlation between each feature in $F_{rgbp}$ with all features in $F_{text}$, helps determine which features in $F_{rgbp}$ are important to determine the

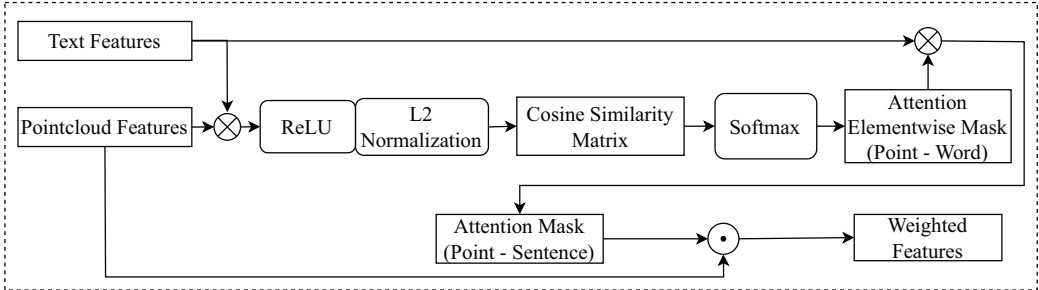

**Figure 4** **The architecture of the attention mechanism with cosine similarity.** $\odot$ denotes element-wise multiplication. $\otimes$ denotes matrix multiplication.

location of the object mentioned in the text. The attention element-wise mask is computed as follows:

$$AEM = Softmax\left[\frac{\delta(F_{rgbp} \cdot (F_{text})^T)}{\|F_{rgbp}\|\|F_{text}\|}\right] \tag{1}$$

where $\delta$ denotes for *ReLU* activation function. And, the attention mask *AM* is computed as follows:

$$AM = \frac{\delta(AEM \cdot (F_{text})^T)}{\|AEM\|\|F_{text}\|}. \tag{2}$$

The weighted features, denoted as $F_{weighted}$, are calculated by element-wise multiplication between $F_{rgbp}$ and *AM*. This produces the final attention weights, which are used to focus the network's attention on certain parts of $F_{rgbp}$.

$$F_{weighted} = F_{rgbp} \odot AM \tag{3}$$

where $\odot$ denotes element-wise multiplication.

### Text-guided grasp generation

This module is refined based on GraspNet (*Fang et al., 2020*) modules including Approach Network, OperationNet, ToleranceNet. Approach Network estimates the approach vectors and possible grasping points, providing M(2 + V) values where M represents the number of chosen grasping points, two represents the graspable binary class or not and V represents the number of approach vectors predefined. OperationNet predicts the operating parameters of the grasp pose, including the orientation and translation of the gripper under the camera frame, and the gripper width. ToleranceNet predicts the grasp robustness, indicating the maximum perturbation that the grasp pose can resist during the grasping process.

### Loss function

The loss function is refined based on the loss function of GraspNet (*Fang et al., 2020*):

$$L = \lambda_1 L_{objectness} + \lambda_2 L_{viewpoint} + \lambda_3 L_{grasp}. \tag{4}$$

The objectness loss $L_{objectness}$ expressed through cross-entropy, calculates the penalty for incorrectly classifying the presence of an object in the point cloud, is given by:

$$L_{objectness} = \sum_{n=1}^{N} \frac{l_n}{N} \tag{5}$$

where $l_n$ given by:

$$l_n = -\sum_{c=1}^{C} w_c \log \frac{\exp(x_{n,c})}{\sum_{i=1}^{C} \exp(x_{n,i})} y_{n,c} \tag{6}$$

where $N$ represents the number of points in the point cloud, $C$ is the number of classes. In the text-guided grasp generation task, when the number of points in the point cloud is large but the number of points containing the object to be grasped is limited, an imbalanced data phenomenon occurs. This leads to the model having a tendency to classify the entire point cloud as not containing the object during training. Therefore, the weight ($w$) in Eq. (6) formula is used to balance the amount of points that contain the object and those that do not.

The viewpoint loss function $L_{viewpoint}$ calculates the penalty for mistakes in predicting the best viewpoint to grasp the object. It uses the Mean Squared Error loss function but only considers predictions for foreground regions and viewpoints classified as good (score above a threshold).

The grasp loss function $L_{grasp}$ is defined as:

$$L_{grasp} = \lambda_4 L_{score} + \lambda_5 L_{rot} + \lambda_6 L_{width} + \lambda_7 L_{tolerance}. \tag{7}$$

The $L_{grasp}$ comprises losses for grasp score loss ($L_{score}$), in-plane rotation classification loss ($L_{rot}$), grasp width regression loss ($L_{width}$), and grasp tolerance regression loss ($L_{tolerance}$). Regression losses employ L1-smooth loss, while classification losses use standard cross-entropy loss.

## EVALUATION

### Evaluation metrics

To judge how well our system predicts grasp poses, especially the most promising ones, we use a metric called *Precision@k* (*Fang et al., 2020*). This metric considers how many of the top k grasp predictions are actually correct for grabbing a specific object. We figure this out by checking the point cloud data within the gripper for each predicted pose and using a force-closure analysis (considering different levels of friction coefficients $\mu$) to see if it would work. In the context of our text-guided approach, *Precision@k* indirectly evaluates vision-language grounding accuracy, as successful grasps require the model to correctly align the textual description with the visual object before generating appropriate grasp poses. A direct vision-language grounding metric, such as text-image cosine similarity or CLIP-based alignment scores, could provide additional insight and is considered for future work.

Since good robot grasp often relies on the very first suggested options, *Precision@k* focuses on the accuracy of those top predictions. We calculate it by ranking all the predicted poses based on their confidence scores, then picking the top k ones, and finally checking how many of those k actually work (true positives).

To get a more complete picture, we calculate *AveragePrecision@μ* ($AP_\mu$) in a range of k values (typically 1 to 50) for different levels of friction coefficients. This $AP_\mu$ metric is especially important for cluttered environments where robots need to prioritize the best grasp options first. In our experiments, we will report $AP_\mu$ results under various conditions, including different settings and friction coefficients $\mu$. This will help demonstrate how well our system finds good 6-DoF grasps in various situations.

## Evaluation on GraspNet-1Billion dataset

The main purpose in this section is to evaluate the overall performance of the model, and we compared it with the current state-of-the-art methods. In general, these models achieve impressive results and make significant contributions to the field. We used the Precision@k metric to provide a comprehensive evaluation of the methods. To ensure fairness, we reimplemented the methods using the source code provided by the authors. The results in Table 2 present the performance on "Seen" objects using the Kinect sensor, showing our improvement with an average precision (*AP*) of 53.2%, $AP_{0.8}$, $AP_{0.6}$, and $AP_{0.4}$ of 63.1%, 55.6%, and 44.1%, respectively. These results demonstrate that our method improves upon other methods. Even when compared with different thresholds ($AP_{0.8}$, $AP_{0.6}$, and $AP_{0.4}$), our method retains superior performance, demonstrating accurate and reliable capture. This performance highlights the advantage of our multimodal approach, which leverages semantic information from text to disambiguate objects, combined with RGB and point cloud data for robust grasp detection. This targeted prioritization is crucial for robots operating in cluttered environments where choosing the best grasp attempt is critical. Additionally, the system operates with exceptional efficiency, processing each grasp prediction in just 260 milliseconds. This rapid processing time makes it suitable for real-time applications where time-sensitive decisions are essential.

The Table 3 aims to demonstrate two important points. First, we show that improving the model's input by incorporating color information in addition to point cloud data is beneficial. Instead of only using point cloud data to extract features, we leverage the geometric information from the point cloud and add color information. This significantly enhances object detection performance. Specifically, from the results table, for the cosine similarity fusion mechanism, the AP scores without color are 45.4%, 52.1%, 46.2%, and 36.3%, while with color the AP scores are 53.2%, 63.1%, 55.6%, and 44.1%. Similarly, other fusion mechanisms also show an approximate 1% improvement when color information is added. Second, we compare three different fusion methods. The first method is multiplication. Although this method allows the model to learn nonlinear relationships between variables, it encounters issues when features have very different ranges. This leads to features with larger values dominating, reducing the influence of other features and causing imbalance in the model. Consequently, the multiplication method may decrease the model's effectiveness in accurately representing object characteristics. The second

**Table 2 The results on GraspNet-1Billion test set captured by kinect sensors.**

| Method | $AP$ | $AP_{0.8}$ | $AP_{0.6}$ | $AP_{0.4}$ |
|---|---|---|---|---|
| GG-CNN (*Morrison, Corke & Leitner, 2018*) | 16.9% | 22.5% | 17.1% | 11.2% |
| *Chu, Xu & Vela (2018)* | 17.6% | 24.7% | 17.9% | 12.7% |
| GPD (*Pas et al., 2017*) | 24.4% | 30.2% | 24.8% | 13.5% |
| PointNetGPD (*Liang et al., 2019*) | 27.6% | 34.2% | 28.3% | 17.8% |
| *Fang et al. (2020)* | 29.9% | 36.2% | 30.6% | 19.3% |
| *Gou et al. (2021)* | 32.1% | 39.5% | 32.5% | 20.9% |
| Contact-GraspNet (*Sundermeyer et al., 2021*) | 31.4% | 31.4% | 31.8% | 21.6% |
| *Zheng et al. (2023)* | 36.1% | 44.0% | 36.6% | 26.0% |
| VoteGrasp (*Hoang, Stork & Stoyanov, 2022*) | 37.5% | 45.6% | 37.6% | 27.6% |
| *Xuan Tan et al. (2024)* | 38.5% | 43.1% | 38.8% | 29.3% |
| Ours | **53.2%** | **63.1%** | **55.6%** | **44.1%** |

**Note:**
The results for our proposed method are highlighted in bold.

method is concatenation. This method is simple and flexible, and does not require complex operations to combine features. However, it does not handle nonlinear relationships between features of the text encoder and visual encoder. The final method is cosine similarity. This method measures the similarity between two vectors without being affected by differences in scale or magnitude of the features. Experimental results indicate that using cosine similarity for fusion between features from the text encoder and visual encoder is the most suitable approach for our study.

## Grasping robot simulation experiment

To assess our method's ability to generate diverse and effective grasps, we conducted simulation experiments in CoppeliaSim (*Rohmer, Singh & Freese, 2013*) using a robotic arm with a multi-fingered gripper. For each experiment, we fed the same object point cloud and different text prompts into the system. The generated grasps were evaluated based on quality, stability, and feasibility before simulation. As Fig. 5 demonstrates, varying text prompts led to distinct grasp configurations for the same object, highlighting the system's capacity to produce task-specific grasps. These results indicate the potential for our approach to enable robots to perform complex manipulation tasks in real world environments. However, extensive validation on physical robotic platforms is necessary to confirm real-world applicability, as discussed in "Limitations".

## IMPLEMENT DETAILS

In our implementation, we utilize the BERT (*Devlin et al., 2019a*) model, as the encoder for text describing objects. The output textual features comprises 256 channels. For point cloud feature extraction, we randomly sampled 20,000 points from depth images and employed a PointNet++ (*Qi et al., 2017*)-based feature learning network, which also yields a 256 channel output. We set $\lambda_1 = \lambda_2 = 1$, $\lambda_3 = 0.2$ and $w_0 = 0.01$, $w_1 = 0.99$, where $w_0$ is the weight for class 0 (no object) and $w_1$ is the weight for class 1 (has object). Our network is trained entirely using a batch size of 4 and optimized with Adam, employing a learning

**Table 3 Results of our network excision study.** See text for details.

| Method | $AP$ | $AP_{0.8}$ | $AP_{0.6}$ | $AP_{0.4}$ |
|---|---|---|---|---|
| Ours (*No Color + Mul*) | 42.3% | 51.5% | 44.8% | 34.4% |
| Ours (*Color + Mul*) | 45.7% | 52.3% | 46.6% | 36.2% |
| Ours (*No Color + Concat*) | 45.3% | 52.9% | 47.2% | 37.3% |
| Ours (*Color + Concat*) | 46.1% | 54.6% | 48.9% | 39.0% |
| Ours (*No Color + CoS*) | 45.4% | 52.1% | 46.2% | 36.3% |
| Ours (*Color + CoS*) | **53.2%** | **63.1%** | **55.6%** | **44.1%** |

Note:
The entry for our best-performing model configuration is highlighted in bold.

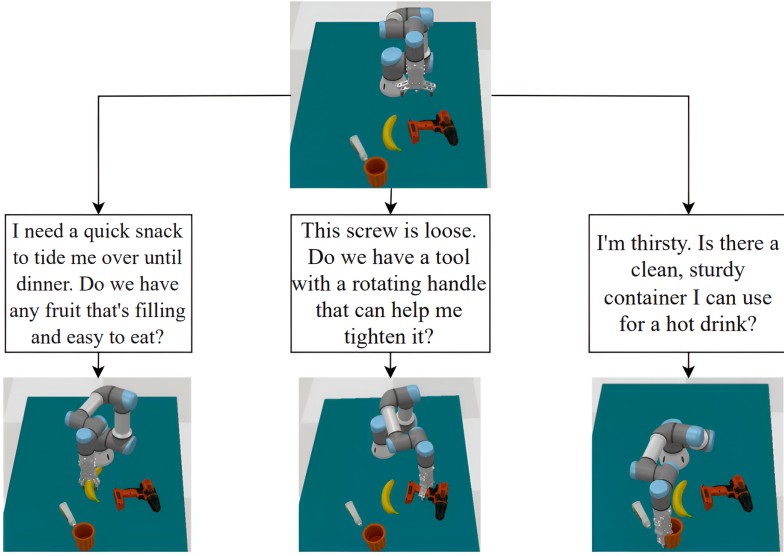

**Figure 5 Example of robot grasp simulation in CoppeliaSim (*Rohmer, Singh & Freese, 2013*).** With the same input point cloud and different text prompts, the system outputs different grasps for the objects described in the text.

rate of 0.001 for 13 epochs. Training on a single Nvidia GeForce RTX 4070Ti 12GB GPU takes approximately 46 h.

## LIMITATIONS

Our approach demonstrates significant advancements, but several limitations remain that warrant further exploration:

**Depth estimation accuracy:** Although Depth Anything V2 (*Yang et al., 2024b*) improves metric accuracy through its teacher-student framework, the generated depth maps may still exhibit a domain gap when applied to uncalibrated real-world scenes with diverse sensor characteristics. Further adaptation, such as fine-tuning in real-world depth data, could enhance robustness.

**Real-world validation:** Our evaluations are primarily based on simulations and the GraspNet-1Billion dataset. Extensive testing on physical robotic platforms across varied

objects and environments is essential to validate real-world performance and address practical challenges such as sensor noise and dynamic lighting.

**Intrinsic parameter sensitivity:** The automated pipeline assumes standard intrinsic parameters from the Kinect depth sensor. Variations in real-world camera intrinsics may require recalibration or adaptation to maintain performance, necessitating further investigation into robust generalization across diverse sensor setups.

**Vision-language grounding metrics:** The Precision@k metric indirectly evaluates vision-language alignment, but direct metrics, such as CLIP-based text-image similarity scores, could provide deeper insight into the model's ability to align textual descriptions with visual objects, particularly in ambiguous scenarios.

**Computational cost:** The dataset generation pipeline, involving Stable Diffusion XL and Depth Anything V2, is computationally intensive. Although this is an upfront cost for creating large-scale training data, optimization strategies could reduce resource demands for greater accessibility.

Future work will focus on addressing these challenges by improving depth estimation realism, conducting real robot experiments, improving sensor generalization, and incorporating explicit vision-language grounding metrics.

## CONCLUSION

This study introduces a novel text-guided grasp detection framework that integrates RGB images, 3D point clouds, and textual descriptions processed by a large language model, achieving state-of-the-art performance with an average precision of 53.2% on the GraspNet-1Billion dataset. By combining geometric, visual, and semantic information, our approach significantly enhances object disambiguation and grasp generation in cluttered environments. We also contributed a text prompt dataset covering 52 object categories from the GraspNet-1Billion dataset, enabling language-driven grasp generation for diverse objects. Additionally, our automated dataset creation pipeline leverages advanced technologies, including Gemini 2.0, Stable Diffusion XL, Depth Anything V2, and GraspNet, to generate diverse and high-quality grasping data with minimal human effort.

Despite these advancements, the model's generalization to novel objects remains limited due to insufficient diversity in object shapes, colors, and textures in the training data. This can lead to reduced performance in real-world scenarios with complex lighting, reflective surfaces, or transparent objects. The reliance on relative depth maps from Depth Anything V2, while improved through its teacher-student framework, still poses challenges in achieving metric accuracy for uncalibrated real-world scenes. Future research will focus on expanding dataset diversity to include a wider range of object characteristics and environmental conditions, leveraging advanced depth estimation techniques like Depth Anything V2 to further bridge the domain gap, and conducting extensive real-robot experiments to validate performance in practical settings. Furthermore, integrating direct vision-language grounding metrics, such as CLIP-based scores, could enhance the model's ability to handle ambiguous instructions, paving the way for more robust and versatile robotic grasping systems.

# ACKNOWLEDGEMENTS

We acknowledge the use of Grok 3, developed by xAI, for correcting English grammar in the final version of the manuscript to ensure clarity and linguistic accuracy.

### Funding

The authors received no funding for this work.

### Competing Interests

The authors declare that they have no competing interests.

### Author Contributions

- Van Duc Vu conceived and designed the experiments, performed the experiments, performed the computation work, prepared figures and/or tables, and approved the final draft.
- Van Thiep Nguyen conceived and designed the experiments, performed the experiments, performed the computation work, prepared figures and/or tables, and approved the final draft.
- Nam Hai Pham analyzed the data, prepared figures and/or tables, and approved the final draft.
- Dinh-Cuong Hoang conceived and designed the experiments, performed the experiments, performed the computation work, authored or reviewed drafts of the article, and approved the final draft.
- Phan Xuan Tan conceived and designed the experiments, analyzed the data, authored or reviewed drafts of the article, and approved the final draft.

### Data Availability

The Vision Data (GraspNet-1Billion Dataset) is available at GraspNet repository and Zenodo at: https://github.com/graspnet. Fang, H. (2025). GraspNet-1Billion Dataset [Data set]. Zenodo. https://doi.org/10.5281/zenodo.16006850.

The Text Data (Object Descriptions for GraspNet Dataset) which describes objects in the GraspNet dataset, was generated by the authors and is available in the Supplemental File and at Zenodo: Van Duc, & Thiep1808. (2024). Text-Guided-RGBP-grasp-Generation: v1.0.0 (v1.0.0). Zenodo. https://doi.org/10.5281/zenodo.14038077.

The code and model checkpoints are also available at Zenodo: Van Duc, & Thiep1808. (2024). Text-Guided-RGBP-grasp-Generation: v1.0.0 (v1.0.0). Zenodo. https://doi.org/10.5281/zenodo.14038077.

### Supplemental Information

Supplemental information for this article can be found online at http://dx.doi.org/10.7717/peerj-cs.3060#supplemental-information.

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
