# Peer review of "Text-guided RGB-P grasp generation"

_PeerJ Computer Science, doi:10.7717/peerj-cs.3060_

## Round 0.1 · original submission · Major Revisions

Thank you for submitting your manuscript. After careful consideration of the reviewers’ reports, I must request a major revision. Although your multimodal RGB–point-cloud framework and automated dataset pipeline are promising, essential implementation details and validation steps are missing or unclear.

Please clarify the precise role of the large language model in the grasping pipeline, expand the related-work coverage, justify the depth-to-point-cloud conversion, add vision-language grounding metrics, and demonstrate generalization on real-world setups.

Kindly revise the paper accordingly and provide a point-by-point response to each reviewer comment; we look forward to reassessing your improved submission.

**Language Note:** The review process has identified that the English language must be improved. PeerJ can provide language editing services - please contact us at [email protected] for pricing (be sure to provide your manuscript number and title). Alternatively, you should make your own arrangements to improve the language quality and provide details in your response letter. – PeerJ Staff

Reviewer 1 ·

Basic reporting

Summary of the paper: This paper presents a multimodal approach to object grasping that integrates 3D point clouds, RGB images, and textual descriptions into a unified RGB-Point Cloud representation. This fusion of data types improves object recognition and grasping accuracy. Additionally, the authors introduce an automated dataset creation pipeline that leverages advanced models (LLMs, Stable Diffusion, Depth Anything, GraspNet) to efficiently generate high-quality datasets with minimal manual annotation.

To improve the paper's quality and clarity, I recommend addressing the following points:
(1) In the related work section, the citation to Xiaocheng Lu (44) should be updated to 'Xiaocheng Lu et al. (44),' as the paper has multiple authors. A similar issue occurs with the citation to Youngjae Yu (19) in Line 184.

(2) Additionally, the related work section is lacking in coverage, particularly with regard to literature on LLM-based robotic grasping. Some relevant works that are missing include:
[1] Reasoning Grasping via Multimodal Large Language Model
[2] Polaris: Open-ended Interactive Robotic Manipulation via Syn2Real Visual Grounding and Large Language Models
[3] GraspGPT: Leveraging Semantic Knowledge from a Large Language Model for Task-Oriented Grasping
[4] A Joint Modeling of Vision-Language-Action for Target-Oriented Grasping in Clutter

Experimental design

(1) Regarding the dataset: In the automated dataset shown in Fig. 1, Depth Anything is used to estimate depth, which is then back-projected into a point cloud. However, there are two concerns not addressed in the paper.
1. The depth information from Depth Anything cannot be directly converted into a 3D point cloud without using carefully selected intrinsic parameters. Without this, the projected point cloud will be distorted. How does the paper address this issue?
2. The depth estimates from the Depth Anything model exhibit a large domain gap compared to real depth data captured from a depth sensor, as seen in the Grasp-1 Billion dataset. How can the method be adapted to work with real-world images in actual robotic applications?

(2) Additionally, the evaluation metric lacks a measure of visual-language grounding accuracy, which is a key contribution of the paper. Instead, it focuses on linguistic-agnostic grasp pose estimation, as shown in Table 2.

Validity of the findings

(1) In the conclusion, the paper claims that the 'generation pipeline based on advanced technologies efficiently generates diverse and realistic capture data.' However, the depth data from Depth Anything significantly differs from real-world depth data captured by an actual depth sensor.

(2) Additionally, the experiments do not assess whether the model trained on the proposed automated datasets generalizes well to real-world robot setups.

(3) The paper lacks a discussion of the limitations of the proposed method. An analysis of these limitations would help provide a better understanding of the method's capabilities.

Cite this review as
Anonymous Reviewer (2025) Peer Review #1 of "Text-guided RGB-P grasp generation (v0.1)". PeerJ Computer Science

·

Basic reporting

The novelty of the multimodal approach that seamlessly integrates 3D data (shape) and RGB images (color, texture) with LLMs should be classified. Study contributions should be indicated clearly.
What is the main contribution? Clarify.
How to integrate LLMs with the proposed model? Clarify.

Experimental design

Please indicate some questions as follows:
1- What are data sources? Explain
2- Which environment was used in the implementation? Open AI / Lamma. How about on the prime Server?
3- The overall proposed model should be designed in this paper.

Validity of the findings

What are evaluation metrics? Clarify.
Indicate the advantages of the proposed models when conducting the experiment.

Additional comments

The English language should be significantly improved.

Cite this review as
Van Pham H (2025) Peer Review #2 of "Text-guided RGB-P grasp generation (v0.1)". PeerJ Computer Science

---

## Round 0.2 · accepted · Accept

Thank you for your careful and thorough revisions. I am pleased to inform you that your manuscript is now accepted for publication.

Both reviewers agree that the updated version has satisfactorily addressed the previously raised concerns. Reviewer 1 commended the improved clarity, detailed methodology, and well-articulated contributions. While they noted that including quantitative results for the Precision@k metric would further strengthen the evaluation, this omission is not critical at this stage. Reviewer 2 also confirmed that all prior issues have been resolved.

Congratulations on your work, and we look forward to its publication.

Reviewer 1 ·

Basic reporting

The revised manuscript enhances sentence clarity and corrects citation formatting in the related work section. It effectively addresses the major concerns I noted in the previous version. and it add more details of the data generation, depth back-projected to point cloud and etc.The proposed method is now presented more clearly, and the authors' contributions are explicitly articulated.

Experimental design

While the paper references the Precision@k metric, Tables 2 and 3 only present results for Average Precision (AP), and no quantitative results of Precision@k is provided. Including these results would strengthen the evaluation.

Validity of the findings

no comment

Additional comments

The updated manuscript has adequately addressed the concerns regarding the details of depth back-projection to point clouds. However, some minor revisions remain necessary, such as including the quantitative results for the Precision@k metric.

Cite this review as
Anonymous Reviewer (2025) Peer Review #1 of "Text-guided RGB-P grasp generation (v0.2)". PeerJ Computer Science

·

Basic reporting

The paper has been revised it.

Experimental design

It is fined

Validity of the findings

The paper has been solved all of the problems

Cite this review as
Van Pham H (2025) Peer Review #2 of "Text-guided RGB-P grasp generation (v0.2)". PeerJ Computer Science